# Hydrocarbon Sorption in Flexible MOFs—Part III: Modulation of Gas Separation Mechanisms

**DOI:** 10.3390/nano14030241

**Published:** 2024-01-23

**Authors:** Hannes Preißler-Kurzhöfer, Marcus Lange, Jens Möllmer, Oliver Erhart, Merten Kobalz, Harald Krautscheid, Roger Gläser

**Affiliations:** 1Institut für Technische Chemie, Fakultät für Chemie und Mineralogie, Universität Leipzig, Linnéstraße 3, D-04103 Leipzig, Germany; 2Institut für Nichtklassische Chemie e.V., Universität Leipzig, Permoserstraße 15, D-04318 Leipzig, Germany; lange@inc.uni-leipzig.de (M.L.); moellmer@inc.uni-leipzig.de (J.M.); 3Institut für Anorganische Chemie, Fakultät für Chemie und Mineralogie, Universität Leipzig, Johannisallee 21, D-04103 Leipzig, Germanykrautscheid@rz.uni-leipzig.de (H.K.)

**Keywords:** metal–organic frameworks, gas mixture separation, flexible materials

## Abstract

Single gas sorption experiments with the C4-hydrocarbons *n*-butane, *iso*-butane, 1-butene and *iso*-butene on the flexible MOFs *Cu-IHMe-pw* and *Cu-IHEt-pw* were carried out with both thermodynamic equilibrium and overall sorption kinetics. Subsequent static binary gas mixture experiments of *n*-butane and *iso*-butane unveil a complex dependence of the overall selectivity on sorption enthalpy, rate of structural transition as well as steric effects. A thermodynamic separation favoring *iso*-butane as well as kinetic separation favoring *n*-butane are possible within *Cu-IHMe-pw* while complete size exclusion of *iso*-butane is achieved in *Cu-IHEt-pw.* This proof-of-concept study shows that the structural flexibility offers additional levers for the precise modulation of the separation mechanisms for complex mixtures with similar chemical and physical properties with real selectivities of >10.

## 1. Introduction

Flexible MOFs or porous coordination polymers represent a unique but large family of crystalline 3D solids built by coordinative bonds between inorganic nodes and organic linkers [1,2,3]. Since their first discovery, an enormous amount of research has been conducted on this material class, utilizing the vast degrees of freedom for the synthesis of new materials, analyzing their properties towards, e.g., gas storage [4,5,6,7], catalysis and even sensor design [8,9] or drug delivery [10,11]. However, given the industrial relevance, energy-efficient gas separation via sorption using MOFs is one of the focus topics within the research community [12,13,14,15,16]. Due to the modularity of the material class, pore geometries are envisioned to be tailored to the specific requirements to achieve good selectivities via size/shape exclusion or kinetic separation as well as for a thermodynamic separation based on differences in adsorbate–surface interactions of at least two concurring gases.

In recent years, remarkable progress was made, e.g., using ultramicroporous MOFs for the separation of propane and propene [15], reaching almost complete exclusion of propane due to size effects and thus a kinetic separation. Another example of the same separation is the MOF ZIF-8, in which the tight pore apertures lead to an enormously decreased diffusivity towards propane as compared to propene [17,18]. In the last two decades, a new subclass of MOFs displaying structurally flexible behavior was found, showing hysteresis in the adsorption isotherm with a large increase in pore volume within a narrow pressure range. Some studies also showcase their potential for separation.

Early work on one of the most prominent flexible MOFs, Mil-53, already showed the potential of this novel material class for CH_4_/CO_2_ gas mixture separations due to the higher affinity towards CO_2_ [19,20,21,22,23]. However, within these studies, it was found that these capabilities are merely based on thermodynamic separation as the species opening the framework (CO_2_) also allows the second, less preferred species (CH_4_) to enter due to the wide pore apertures in the MOF. Similar observations were also made by other groups for other materials [18,24,25,26]. While several groups focused on the ability of flexible MOFs to chemically separate very different adsorptives like N_2_, CH_4_ and CO_2_ or specific alkynes [26,27,28,29,30], only a small number of publications deal with the separation of hydrocarbons on flexible MOFs. Van den Bergh et al. analyzed the flexible ZIF-7, a zeolite imidazolate framework, for alkane–alkene separation for C2 and C3 hydrocarbons. It was shown for the first time that flexible MOFs have the ability to separate entirely within breakthrough curve experiments, herein based on the ability to find optimal sites within the pore apertures [31]. On the same material, Chen et al. conducted detailed breakthrough curve analysis for the separation of ethane and ethylene, further verifying the potential of the material class while also proving that experimental results can be sufficiently simulated via DFT calculations [23]. Couck et al. investigated the MOF COMOC-2 for separation of ethane and propane via breakthrough curve experiments with propane selectivity of 2–3 due to the higher affinity for adsorption on the material [32]. The MOFs NJU-Bai8 and NKU-FlexMOF-1 were investigated by Krishna’s group for the separation of propane/propene gas mixtures [33,34] with both showing promising results by experimental breakthrough curves based on a thermodynamic separation mechanism. Cui et al. utilized the ultramicroporous materials MnINA and CuINA, which display flexible behavior, and showed a molecular sieving effect that prefers *n*-butane over *iso*-butane via experimental breakthrough curves [35].

Generally, high focus in the literature is placed on the impact of modifications of the linker or the SBU on the thermodynmic equilibrium but less so on the overall sorption kinetics or potential separation mechanism [35,36,37,38,39]. Furthermore, only a few studies have contributed to a consistent view on the specific kinetics of structural transitions in flexible materials to date with a range of different MOFs and sorptives [40,41,42,43]. The possibility for kinetic separations of similar hydrocarbons, even up to practical size exclusion, has only been shown by Cui et al. [35] so far and should be investigated in more detail.

Therefore, the goal of this study is to investigate the governing thermodynamic and kinetic factors for gas separations on flexible MOFs by the individual study of them for a set of MOFs and different hydrocarbon sorptives. The subsequent analysis of static gas mixture experiments allows deeper insights into the molecular mechanism than break-through curves or theoretical assumptions on sorption isotherms only. Furthermore, key indicators for competitive gas separations on flexible materials will be derived in order to promote future research on the topic.

As probe molecules, the C4-hydrocarbons *n*-butane, *iso*-butane, 1-butene and *iso*-butene were used, enabling the analysis of both olefins and paraffins with different spatial demands (kinetical diameters of 4.32–4.72 Å). The gases were probed on the MOFs [Cu_2_(H-Me-trz-Ia)_2_] and [Cu_2_(H-Et-trz-Ia)_2_], with two isoreticular frameworks which are hereafter referred to as *Cu-IHMe-pw* and *Cu-IHEt-pw* for simplicity. These MOF systems have been previously investigated for their thermodynamic [44] and kinetic properties [45]. The linkers are based on triazolyl-isophthalate and only deviate in the alkyl side-chain at the 2-position of the triazol ring. The deviating names thus refer to a methyl and ethyl group, respectively. This further allows the investigation of the impact of the linker size on the separation mechanism. The bridging coordination of the carboxylate groups of the linkers results in a square planar CuO_4_ environment of the metal ions, leading to the well-known dinuclear paddle-wheel motif. Through coordination of a nitrogen atom of the triazolyl group in the apical positions of the metal centers, a three-dimensional network is assembled. Crystallographic data and CO_2_ adsorption isotherms were reported by Kobalz et al. [46]. *Cu-IHMe-pw* exists in three stable phases with two distinct gate-openings while *Cu-IHEt-pw* exhibits one structural transition only. Crystal structure details for all three phases of *Cu-IHMe-pw,* denoted as narrow pore (*np*-phase), medium pore (*mp*-phase) and large pore phase (*lp*-phase), were resolved by single-crystal and powder X-ray diffraction experiments. Based on these experiments, *Cu-IHMe-pw* was shown to have a pore size of around 3–5 Å [46] and thus is within the range of the kinetic diameters of the chosen adsorptives of 4.3 to 5.3 Å [47]. The corresponding crystal structure data of the *np*- and *mp*-phases are shown in the Appendix A.

## 2. Materials and Methods

### 2.1. Synthesis of MOFs

The metal–organic frameworks [Cu_2_(H-Me-trz-Ia)_2_] and [Cu_2_(H-Et-trz-Ia)_2_] (herein called *Cu-IHMe-pw* and *Cu-IHEt-pw*, respectively, for simplicity) were synthesized according to the original procedure reported elsewhere [46].

### 2.2. Measurement of Equilibrium Sorption Isotherms

The adsorption and desorption isotherms of the C4-hydrocarbons *n*-butane, *iso*-butane, 1-butene and *iso*-butene on the MOF systems were measured in a temperature range from 283 K to 313 K and at pressures up to 300 kPa using a magnetic suspension balance (Fa. Rubotherm GmbH, Bochum, Germany). Three pressure transducers (MKS Instruments Deutschland GmbH, Munich, Germany, Omega Engineering GmbH, Deckenpfronn, Germany) were used to gather accurate data for the whole pressure range up to 300 kPa. In the preparation of the sorption experiments, typically a stainless-steel sample holder was filled with around 0.2 g of a MOF sample. The sample cell was evacuated for at least 12 h at 373 K and a minimum pressure of 0.3 Pa was applied until constant mass was achieved. Subsequently, the gas was dosed into the balance and the pressure was increased stepwise after reaching the equilibrium. Sorption equilibrium was assumed to be reached when no further weight increase and pressure change of less than 1 Pa within 15 min were observed. The temperature was kept constant with an accuracy of 0.5 K. The gases were purchased from Linde (Linde AG, Munich, Germany) with purities of 99.5%. In order to calculate the surface excess mass from the measured weight values, a buoyancy correction was carried out. Furthermore, absolute gas loadings were calculated. Detailed descriptions for these procedures can be found elsewhere [48]. The densities for each gas were calculated with the program FLUIDCAL [49].

### 2.3. Measurement of Static Gas Mixtures

The static sorption equilibria of gas mixtures were determined by means of a hybrid manometric–gravimetric system built by Lange [50], the schematic structure is shown in Appendix A. Herein, the sample chamber of the magnetic suspension balance was integrated into a manometric arrangement. The manometric part consists of pressure vessels and their piping. The modular design of the apparatus allows three distinct take-aways: First, the gravimetric measurements allow the precise mass calculation of the adsorbent during both activation as well as gas uptake. Second, by deploying mass balance calculations for the manometry, the partial molar loadings of both components on the adsorbent can be calculated. This requires knowledge of the gas phase composition, which was determined externally in a gas chromatograph (GC) with a flame ionization detector (FID). A more detailed description of the approach is provided in the Appendix A.

## 3. Results

### 3.1. Single-Gas Thermodynamic Analysis

For the analysis of the equilibrium sorption isotherms of the studied systems, the Dubinin approach [51,52] is used as in previous studies for the MOF systems under investigation [44,45]. It has the advantage of enabling the analysis of sorption equilibria at different temperatures at once, enabling an easy and quick identification of key differences in sorption equilibria for different sorptive–sorbent systems. Figure 1 presents the band of isotherms for the ad- and desorption of the studied C4-hydrocarbons on *Cu-IHMe-pw* and *Cu-IHEt-pw* for the temperatures 283, 298 and 313 K in a characteristic Dubinin plot, each showing the specific pore volume W in dependence of the sorption potential A. Furthermore, every resulting characteristic curve for each pair of adsorbent and adsorptive as well as adsorption and desorption was fit with a dual-volume Dubinin–Asthakov approach, as was carried out previously [44,45]. These fits build the basis for the analysis in the following. A more detailed description of the fitting equation and methodology is given in the ESI, Appendix A.

Overall, the measured points of sorption converge into mostly sharp characteristic curves for every single adsorptive–adsorbent system. Three deviations from the expected patterns are observable within the MOF systems. *Iso*-butane is very slow to open the framework in *Cu-IHMe-pw* (points in Figure 1 upper left do not resemble equilibrium) and it is not able at all to open *Cu-IHEt-pw* within a reasonable timeframe (<1 week), indicating that the visualized datapoints do not represent equilibrium states. Thus, this phenomenon is of a kinetic nature and analyzed in more detail within the next section. Furthermore, *n*-butane does not show a distinct gate-closing within *Cu-IHEt-pw,* which is an outlier in the dataset and likely due to steric blocking effects of the adsorptive within the framework.

The specific boundaries of the structural transition, meaning the gate-opening end $A^{GOE}$ for adsorption and the gate-closing start $A^{GCS}$ for desorption, can be determined by applying the ESW theory as described by Adolphs [53] and previous work [44]. All data regarding the gate-opening end and gate-closing start points as sorption potentials as well as the corresponding pressures at 298 K are displayed in Table 1.

Due to the flexibility of the frameworks, the characteristic ad- and desorption curves show two distinct sorption regimes: Low sorptive loading within the *np*-phase and high sorptive loading within the *mp*-phase of the respective MOF. Furthermore, the sorption patterns show a distinct hysteresis for all the sorptive–sorbent systems studied, which is common for flexible MOFs [1,54,55].

In a previous study, the sorption thermodynamics in both *Cu-IHMe-pw* and *Cu-IHEt-pw* were investigated. It was concluded that the desorption pattern resembles the actual thermodynamic equilibrium more closely [44], an interpretation that is in line with other authors [56,57]. Therefore, the desorption patterns are taken for the thermodynamic analysis of the interaction potentials and energy differences between the two phases.

When the adsorption potential tends to zero, the specific pore volume of the solid occupied by the fluid can be derived at the point of intersection with the y-axis (Figure 1). From *Cu-IHMe-pw* to *Cu-IHEt-pw*, the specific pore volume is reduced by one third due to the larger linker size and the resulting smaller pore in the opened *mp*-form. Regarding *Cu-IHMe-pw* individually, the accessible pore volume shows a preference firstly towards the branched hydrocarbons and secondly to the paraffins (overall order *iso*-butane > *iso*-butene > *n*-butane > 1-butene). This is likely due to steric reasons as the more compact hydrocarbons may find a denser packing within the opened pores. For *Cu-IHEt-pw*, the overall loadings of *n*-butane, *iso*-butene and 1-butene are almost equal, showing no specific steric preferences while *iso*-butane is not able to open the framework.

The relative positions of the desorption patterns can give insights into the interaction potentials between the different sorptive–sorbent systems. Generally, the further the pattern is shifted to higher sorption potentials, the higher the specific sorption enthalpy and thus the interaction between guest and host. Herein, both olefins show a much higher interaction potential towards both *Cu-IHMe-pw* and *Cu-IHEt-pw* as compared to the paraffins. This is likely due to a stronger interaction of their diffused π-orbitals with the polarizing surfaces. Both the higher affinity and denser packing of olefins in MOFs are commonly observed in the literature and a key reason why this material class is considered as promising separation material [58,59,60,61].

A recently published method to normalize gas properties and to subsequently enable the calculation of normalized interaction potentials called *D-UAT* can be furthermore theoretically confirmed [44]. The reduced interaction potentials of the eight different systems under study are shown in the ESI, emphasizing the higher affinity of both 1-butene and *iso*-butene towards the MOFs even more (e.g., on *Cu-IHEt-pw*, *iso*-butene 103 J mol^−1^ K^−1^, *n*-butane 91 J mol^−1^ K^−1^ at half coverage). In order to derive the total energy difference between the *np*- and *mp*-phases within the MOFs ($∆F^{Host}$), the same approach as utilized in a previous work [44] based on a method by Coudert et al. [62] was applied. Overall, the values of $∆F^{Host}$ are very consistent within the MOF systems with 17.1 and 19.5 J mol^−1^ for *Cu-IHMe-pw* and *Cu-IHEt-pw*, respectively, and in the same range as calculated within Ref. [44].

### 3.2. Single-Gas Kinetic Analysis

In a further analysis step, the individual uptake curves of each pressure increase during the isotherm recordings were derived. These were investigated by means of uptake fitting with an LDF approach [63] and, subsequently, effective transport diffusivities via a simplified methodology were derived as was carried out in a previous work of the group [45]. The diffusivities in dependence of the sorptive loading show three distinct regions as can be seen in Figure 2. Fast uptakes were recorded within the bare *np*- as well as *mp*-phases, leading to diffusivities of around 5∙10^−13^ m^2^ s^−1^ and above, which is comparable to other measurements for, e.g., *n*-butane in microporous solids [64]. However, much slower uptakes are recorded during the structural transition, dropping the diffusivity to values of 1∙10^−14^ and 1∙10^−16^ m^2^ s^−1^ for *n*-butane on *Cu-IHMe-pw* and *Cu-IHEt-pw*, respectively, further exemplifying the kinetic hindrance of the overall process [43,57,65].

Furthermore, the differences between the two MOFs are significant, most likely caused by the tighter pore space of *Cu-IHEt-pw* which may slow down the diffusion to the active sites as well as hinder the molecular reorientation of the framework itself.

Within the different adsorptives, it becomes evident that *n*-butane does open the framework slower as compared to both olefins within both MOFs. While no big difference between the olefins is recognizable in *Cu-IHMe-pw*, *iso*-butene seems to trigger the overall process the fastest of all adsorptives within *Cu-IHEt-pw*. An evaluation of the transport diffusivities of *iso*-butane was not possible as the process is too time-intensive to ensure a proper measurement in either MOF. This is exemplified by the uptake curves for a larger pressure step on *Cu-IHMe-pw* as shown in Figure 3. Herein, the uptake curves for pressure jumps up to 20 kPa and thus beyond the respective gate-opening points were recorded. The *iso*-butane uptake takes more than 1000 times longer as compared to *n*-butane, while both olefin uptakes are completed much quicker.

In a previous study of the system *n*-butane/*Cu-IHMe-pw* [45] and a recent publication by Miura et al. [43], it was concluded that the overall rate of structural transition is dependent on the difference of the sorption potential *A* at the gate-opening pressure and the pressure point being set. Herein, the points of the gate-closing starts (GCSs) from the desorption patterns are taken as reference points and the corresponding potential is set as the minimum potential that has to be overshot to ensure a nearly complete structural transition. The calculated potential difference for each adsorptive can be defined as the driving force of the structural transition as a guide (please see Appendix A for further detail).

However, given the desorption patterns of *n*- and *iso*-butane, the latter has a larger potential difference for the described pressure step but has a much slower rate of structural transition. This effect can likely be ascribed to steric effects of *iso*-butane within the framework, with two potential explanations that have to be further analyzed. As evident by the equilirbium data, *iso*-butane can enter the opened framework but the diffusion through the opened pores might be hampered due to the larger spatial demand of the sorptive with 5.3 Å.

However, it is also possible that the re-orientation of the framework is sterically blocked by the sorptive, leading to a very high framework energy of the intermediate state. This activation energy within the free energy profile would make the overall transition become less likely and thus requires more time to complete. While both hypotheses may hold true, a more in-depth explanation can be found within the Appendix A. Thus, there stands the question whether these kinetic observations can be utilized for potential gas separations.

### 3.3. Binary Gas Mixture—Static Measurements

Within this section, the general applicability of *Cu-IHMe-pw* and *Cu-IHEt-pw* for the separation of binary C4-hydrocarbons is investigated. By utilizing a novel hybrid manometric–gravimetric apparatus set-up [50], it is possible to analyze both the overall kinetic gas uptake as well as the gas composition within the sample chamber in dependence of time. This allows the precise investigation of the complex interplay of thermodynamic and kinetic relations between both gases and the host under static conditions, which has not been conducted so far to the best of our knowledge. The investigation was conducted as follows:(1)The overall gas uptakes recorded via the gravimetric suspension balance of the gas mixture on the MOFs were fit with the aid of the kinetic gate-opening model (“GO” model) by Tanaka et al. [66].(2)A coupled mass balance incorporating the pre-determined overall available gas volume as well as the gas composition monitored during the experiment via gas chromatography enables the calculation of the mass of the adsorbed phase per gas species.(3)An additional fit for the uptake of the slower gas species allows the detailed modeling of the gas phase composition over time as well and thus the evaluation of the selectivity over the whole measurement time.

The overall modeling approach is explained in more detail within the Appendix A. Given the idiosyncrasies of *iso*-butane, the paraffin separation of *n*-butane and *iso*-butane is examined first. For this experiment, a molar 50:50 mix of *n*-butane and *iso*-butane was released to the sample chamber leading to a final pressure of 40 kPa after adsorption. The subsequent partial pressures of 20 kPa each are identical to the single-gas uptakes presented in Figure 3, thus both are overshooting the minimum necessary pressure for a complete gate-opening considerably with 0.6 and 0.5 kPa, respectively. The resulting kinetic gas uptakes, the gas phase composition and separation selectivity are shown in Figure 4.

Herein, it becomes evident that the overall gas uptake is slower for the gas mixture as compared to the single-gas uptake of *n*-butane with the same partial pressure step, although much faster than compared to the bare *iso*-butane adsorption (half coverage is reached after 40 min for the mixture, 1 min for *n*-butane and ~1000 min for *iso*-butane individually). From the evolution of the gas phase composition, it can be seen that *n*-butane is predominantly adsorbed at the beginning, reducing the total gas phase fraction to about 33%, resulting in a total separation factor of a maximum of 10 (at time 200 min). Beyond that, *iso*-butane does continously enter the opened framework, exchanges the adsorbed *n*-butane and incorporates itself within the framework. This leads to a subsequent increase in the gas phase fraction of *n*-butane and a final separation factor after 10,000 min (~7 days) of 0.9 for *n*-butane, meaning *iso*-butane is slightly preferred.

It can be concluded that, although *iso*-butane is preferentially adsorbed under equilibrium conditions (see thermodynamic section), *n*-butane is predominantly adsorbed first and initiates the structural transition. Thus, *n*-butane simultaneously forces a penetration on its part into the first opened pore regions of the *mp*-phase and occupies them immediately, indicating a cascading effect that continuously discriminates the slower species. This is the first time that such a complex interplay of thermodynamic and kinetic separation mechanisms has been observed within flexible MOFs to the best of our knowledge. However, as the maximum separation factor reaches 10, a complete exclusion of *iso*-butane is not observed. This is probably due to the preferential adsorption of *iso*-butane in the pore entries of the particles. The consequent slow re-exchange also shows that the pores are wide enough to enable such a counter diffusion. It can therefore be concluded that the framework offers both the potential for a thermodynamic separation as well as a kinetic separation, whilst the structural flexibility offers the fine-tuning of both.

Based on these results, it needs to be clarified whether the effect of kinetic preference could be even more harnessed utilizing *Cu-IHEt-pw* with its tighter pore spaces. Therefore, a similar experiment was conducted for the same mixtures albeit with the latter MOF. Herein, the overall pressure step was set to 0–200 kPa, a pressure that largely overshoots the minimum pressure necessary to open the framework for *n*-butane to ensure a complete uptake within an acceptable timeframe. The results can be seen in Figure 5 (please note that the overall sample masses between the experiments presented in Figure 4 and Figure 5 deviate and, thus, the bare gas phase compositions cannot be compared).

Overall, similar observations can be made as for the previous experiment. The sorptive *n*-butane is overall kinetically preferred within this binary mixture, leading to a separation factor of around 11. However, the re-exchange of *iso*-butane cannot be observed even after around 5000 min. This is likely caused by the tighter pore space within *Cu-IHEt-pw* and the subsequent kinetic hindrance of the exchange.

This clearly shows that the overall separation mechanism switches herein from a kinetic separation towards a size exclusion effect and the overall dependence on the linker size. An additional experimental set-up includes a mixture of *iso*-butane and *iso*-butene. The latter is both thermodynamically and kinetically preferred considering single-gas equilibrium and uptake data. The gas composition and the subsequent separation factor are displayed in Figure 6. Herein, an almost complete exclusion of *iso*-butane can be observed, leading to a separation factor of above 50 since the adsorption sites on the pore entries and particle surface also seem to have a higher affinity towards *iso*-butene.

In order to show that a sufficient separation of a gas mixture with much similar thermodynamic affinities and rates of sorption, a 50:50 mixture of 1-butene and *iso*-butene is shown in Appendix A. Herein, a separation factor of 4–5 was reached without a subsequent re-exchange of gases, which can likely also be ascribed to a hybrid thermodynamic as well as kinetic separation pathway with iso-butene having both higher sorption enthalpy and a higher rate of structural transition.

However, there stands the question to what degree the latter is dependent on the former. A high sorption enthalpy per mole of adsorbate may trigger a faster provision of energy to close the energetic gap between both structures. Additional parameters like the framework energy of the intermediate state as well as size exclusion effects may play a pivotal role in the precise modulation of separation mechanisms. These thermodynamic relations are intrinsically dependent on the precise energy profile of the host–guest system. Although computationally very expensive, the calculations of such profiles are major focus topics of the MOF community now [67,68,69]. A subsequent calculation of the rate of structural transition via the transition state theory (TST, more detailed description within Appendix A) may be possible in similar ways as conducted by Camp et al. [70], which would subsequently allow a complete in silico screening of the separation ability of flexible porous solids. Indeed, computational studies concerning large-scale screenings of a vast number of potential porous solids have shown to be very promising, taking into account pore geometries as well as molecular interactions [71,72,73,74,75].

## 4. Conclusions

Within this study, the potential of flexible MOFs for gas separations was investigated by analyzing the governing thermodynamic and kinetic properties individually during static gas mixture experiments. The model adsorbents *Cu-IHMe-pw* and *Cu-IHEt-pw* both show flexible behavior, with the latter having higher equilibrium pressures for the gate-opening and -closing of the structural transition than the former. This is due to a slightly higher energy difference between the two concurring structural conformations (*np*- and *mp*-phase). Probing both frameworks with C4-hydrocarbons in single-gas experiments individually shows a strong preference for the olefins while experiments with *iso*-butane inidicate steric effects with slow diffusion or even size exclusion due to the tight pore widths. To gain deeper insights into the precise interplay of sorption enthalpy and rate of structural transition in dependence of the pressure step, computational methods could be applied in the future. Recent advances in the development of free-energy profiles coupled with the transition state theory could be harnessed in order to allow for large-scale screenings of MOFs for specific separation purposes.

The proof of concept presented here shows that the structural flexibility provides an additional tool for the design of selective adsorbents and the modulation of the separation mechanism from thermodynamic to kinetic or even size exclusion with the alteration of the linker size and pore width. Further investigations will focus on the influence of different pressure steps as well as breakthrough curve analysis in order to verify the effective potential regarding olefin/paraffin separation.

## Figures and Tables

**Figure 1 nanomaterials-14-00241-f001:**
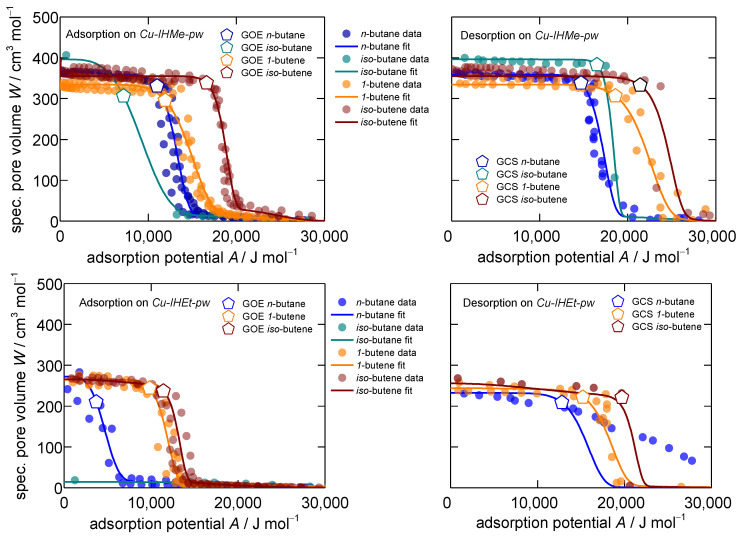
Characteristic patterns for adsorption (**left**) and desorption (**right**) of the C4-hydrocarbons *n*-butane, *iso*-butane, 1-butene and *iso*-butene on *Cu-IHMe-pw* (**top**) and *Cu-IHEt-pw* (**bottom**) measured at temperatures of 283, 298 and 313 K. The gate-opening end (GOE) and gate-closing start (GCS) potentials are individually marked. All isotherm patterns are modeled with the Dubinin–Asthakov approach [51,52]. Please note two idiosyncrasies here: (1) The adsorption from *iso*-butane at *Cu-IHMe-pw* (**top left**) could not be entirely conducted due to very slow uptakes and thus, only one data-point is provided. (2) The desorption fit of *n*-butane on *Cu-IHEt-pw* (**bottom right**) was derived from a previous work based on the *D-UAT* and the desorption of propane on the same material [44].

**Figure 2 nanomaterials-14-00241-f002:**
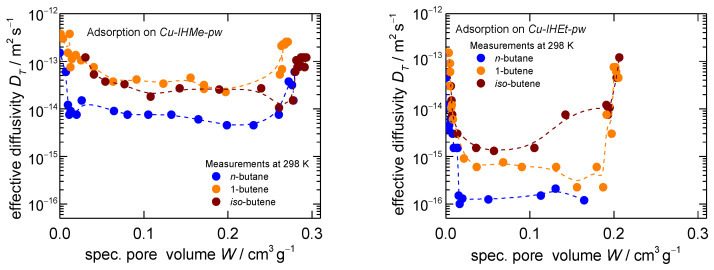
Transport diffusivities calculated from the individual pressure steps during the gravimetric adsorption measurements for *n*-butane, 1-butene and *iso*-butene at 298 K on *Cu-IHMe-pw* (**left**) and *Cu-IHEt-pw* (**right**). Please note that the adsorption for *iso*-butane was too slow to be evaluated in both MOFs.

**Figure 3 nanomaterials-14-00241-f003:**
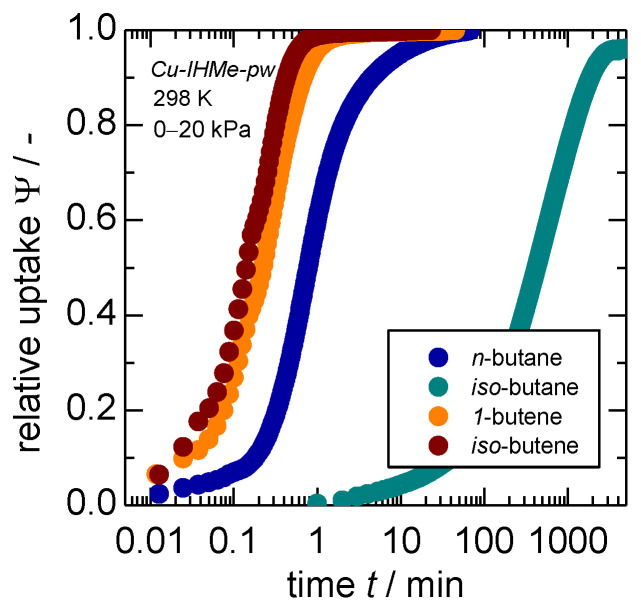
Individual gas uptakes of C4-hydrocarbons on *Cu-IHMe-pw* at 298 K for pressure jumps of 0–20 kPa. Please note that due to the slow uptake of *iso*-butane, another gravimetric set-up was used with a temporal resolution of 1 min as compared to the other measurements with a resolution of 0.77 s.

**Figure 4 nanomaterials-14-00241-f004:**
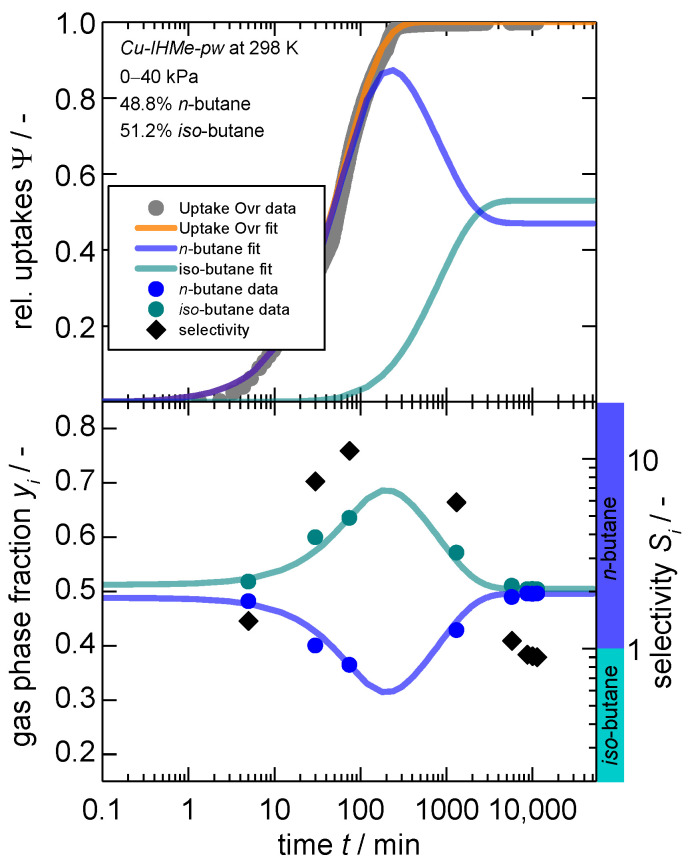
Kinetic gas uptake (**top**), gas phase composition (**bottom, left axis**) and separation selectivity (**bottom, right axis**) in dependence of time for a “50:50” mixture^1^ of *n*-butane and *iso*-butane on *Cu-IHMe-pw* for a pressure jump of 0–40 kPa. Additionally, the overall gas phase compositions and sorption kinetics are modeled via the “GO” model [66] and a mass balance. Within the bottom graph, a selectivity larger than 1 indicates a preference for *n*-butane as indicated by the additional colored ribbons. 1. Please note that a 50:50 molar mixture was aimed for, but actual results show slight deviations with 48.8:51.2.

**Figure 5 nanomaterials-14-00241-f005:**
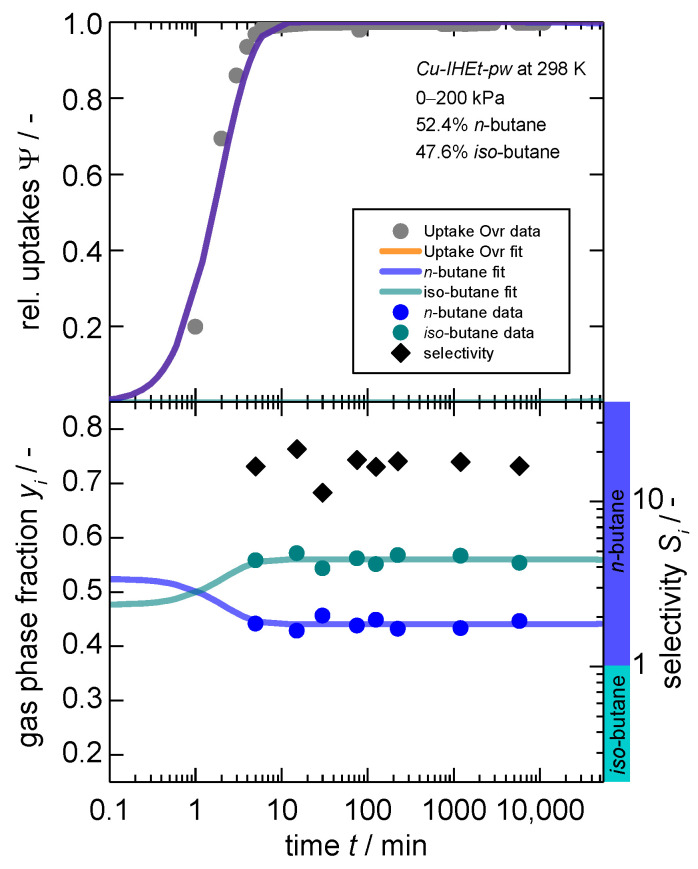
Kinetic gas uptake (**top**), gas phase composition (**bottom, left axis**) and separation selectivity (**bottom, right axis**) in dependence of time for a “50:50” mixture^1^ of *n*-butane and *iso*-butane on *Cu-IHEt-pw* for a pressure jump of 0–200 kPa. Within the bottom graph, a selectivity larger than 1 indicates a preference for *n*-butane as indicated by the additional colored ribbons. 1. Please note two observations: (1) the fit for the overall uptake as well as for *n*-butane are overlapping as almost no *iso*-butane is adsorbed. (2) A 50:50 molar mixture was aimed for, but actual results show slight deviations with 52.4:47.6.

**Figure 6 nanomaterials-14-00241-f006:**
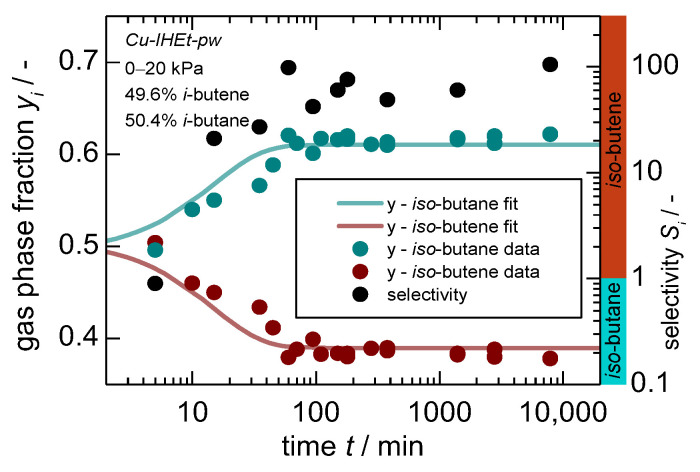
Gas phase composition (left axis) and separation factor (right axis) in dependence of time for a “50:50” mixture^1^ of *iso*-butene and *iso*-butane on *Cu-IHEt-pw* for a pressure jump of 0–20 kPa. Within the bottom graph, a selectivity larger than 1 indicates a preference for *iso*-butene as indicated by the additional colored ribbons. 1. Please note that a 50:50 molar mixture was aimed for but actual results show slight deviations with 49.6:50.4.

**Table 1 nanomaterials-14-00241-t001:** Sorption potentials and pressures at 298 K for all gate-opening end (GOE) and gate-closing start (GCS) points for the C4 adsorption on *Cu-IHMe-pw* and *Cu-IHEt-pw.*

MOF	Adsorptive	$A^{GOE}$ $\;\mathrm{in}\;kJ\;{mol}^{-1}$	$p^{GOE}$ in kPa	$A^{GCS}$ $\;\mathrm{in}\;kJ\;{mol}^{-1}$	$p^{GCS}$ in kPa
*Cu-IHMe-pw*	*n*-butane	11,000	2.9	14,700	0.6
	*iso*-butane	7200	19.2	16,500	0.5
	1-butene	12,000	2.4	18,700	0.2
	*iso*-butene	16,600	0.4	21,400	0.1
*Cu-IHEt-pw*	*n*-butane	3700	54.3	12,800	1.4
	*iso*-butane	-	-	-	-
	1-butene	9900	5.6	15,300	0.6
	*iso*-butene	11,400	3.0	19,700	0.1

## Data Availability

The datasets generated during and/or analyzed during the current study are available from the corresponding author.

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
