# Peer review of "Hydrocarbon Sorption in Flexible MOFs—Part III: Modulation of Gas Separation Mechanisms"

_nanomaterials, 2024, doi:10.3390/nano14030241_

Round 1

Reviewer 1 Report

Comments and Suggestions for Authors

This work conducted C4-hydrocarbons adsorption experiments on the flexible MOFs, and investigated the thermodynamic equilibrium and sorption kinetics. This is quite complex mechanism, in which molecular affinity, structural transition as well as steric effects are involved. The binary gas mixture static measurements provide valuable information to uncover the proposal; while how about under flow feed condition? The dynamic breakthrough performance? Another issue is the readability. For instance, Fig. 1 is not direct to follow. The adsorption isotherm should be given, at least, the typical one that used for the transformation should be provided. Overall, this work is interesting and valuable, I would like to recommend its publication upon considering the above issues.

Comments on the Quality of English Language

many times "gate opening end" and "gate closing start"; better giving one schematic to guide readers

Author Response

In Response to Reviewer #1

Comment 1:

This work conducted C4-hydrocarbons adsorption experiments on the flexible MOFs, and investigated the thermodynamic equilibrium and sorption kinetics. This is quite complex mechanism, in which molecular affinity, structural transition as well as steric effects are involved. The binary gas mixture static measurements provide valuable information to uncover the proposal; while how about under flow feed condition? The dynamic breakthrough performance?

Response and action taken:

We thank the reviewer for his/her comments. We absolutely agree in that breakthrough experiments under flow feed conditions are resembling real industrial applications much more than static measurements. Therefore, we also see this as one of our major focus points going forward. However, we chose static experiments to ascertain that we achieve sorption (and structural) equilibrium. Also, the challenge within the analysis of breakthrough curves lies in the added variables (particle sizes, gas concentration in inert-feed gas, flow rate etc.), that make an in-depth understanding of the underlying mechanism quite challenging. To provide first insights from our first breakthrough experiments: the gases will break through immediately as the MOFs will still be closed and only after some time will start to adsorb gases, which make the curves highly complex. Therefore, in this study, we focused on the bare mechanisms at play first and will look more towards the dynamic experiments in the future. However, we have a further comment within the main manuscript to differentiate our approach (static) to the dynamic experiments.

Comment 2:

Another issue is the readability. For instance, Fig. 1 is not direct to follow. The adsorption isotherm should be given, at least, the typical one that used for the transformation should be provided. Overall, this work is interesting and valuable, I would like to recommend its publication upon considering the above issues.

Response and action taken:

We thank the reviewer for his/her comments herein as well. Fig.1 uses the well established Dubinin-representation of isotherm data, something that we see at least equally valid compared to the classic representation with molar loading vs. pressure. In our view, the Dubinin-approach provides several advantages over the normalization of the loading in terms of volume rather than molar amount as well as the added benefit of further normalizing the applied sorption potential in terms of the reduced sorption potential – essentially reducing differences of characteristic patterns in the Dubinin-representation to mainly host-guest interaction potential differences. Therefore, we stay with the provided Dubinin-approach.

Comment 3:

Many times "gate opening end" and "gate closing start"; better giving one schematic to guide readers

Response and action taken:

We thank the reviewer for his/her comments. The mentioned points serve as reference points within the isotherm analysis, both are related to the structural transition. Herein, the gate-opening end refers to the adsorption while the gate-closing start refers to the desorption. Adsorption and desorption show a significant hysteresis due to the structural transition and therefore, it is not possible to provide only one metric as both are relevant in our opinion. The adsorption curves show usually indicate a kinetic hindrance as these are not as sharp as the desorption curves, with the latter most likely representing the thermodynamic equilibrium much more. Using this hypothesis, one can derive from Fig. 1 that Iso-butane has the largest kinetic hindrance of all analysed sorptives while under thermodynamic equilibrium, it has the second lowest sorption potential (second highest pressure) in which it collapses again. Therefore, we stay with the differentiation herein.

Reviewer 2 Report

Comments and Suggestions for Authors

This work offered by Roger Gläser and co-workers is some interesting. In terms of the contents, this work is well-organized and suitable for direct acceptance. However, due to that the metal-organic frameworks [Cu2(H-Me-trz-Ia)2] and [Cu2(H-Et-trz-Ia)2] (herein called Cu-IHMe-pw and Cu-IHEt-pw respectively for simplicity) were synthesized according to the original procedure reported elsewhere, I strongly suggest that authors carry out the computation to support the description of gas separations, not in future.

Author Response

In Response to Reviewer #2

Comments:

This work offered by Roger Gläser and co-workers is some interesting. In terms of the contents, this work is well-organized and suitable for direct acceptance. However, due to that the metal-organic frameworks [Cu2(H-Me-trz-Ia)2] and [Cu2(H-Et-trz-Ia)2] (herein called Cu-IHMe-pw and Cu-IHEt-pw respectively for simplicity) were synthesized according to the original procedure reported elsewhere, I strongly suggest that authors carry out the computation to support the description of gas separations, not in future.

Response and action taken:

We thank the reviewer for his/her comments. We assume that the comment regarding the computation refers to a statement in our manuscript: “To gain deeper insights into the precise interplay of sorption enthalpy and rate of structural transition in dependence of the pressure step, computational methods could be applied in the future” (to be found in the conclusion section). We definitely agree that this should be a major focus of future work in order to allow for large-scale screenings, similar to the work of Randall Snurr. It is also correct that within our group, the crystal structures are digitally available for Cu-IHMe-pw in both closed and opened phases. However, we see this as a highly complex undertaking that requires a dedicated publication in itself. The mechanisms at play are both thermodynamically as well as kinetically driven and thus, both e.g. Monte-Carlo/DFT as well as Molecular Dynamics studies for both closed and opened phases would be required in order to gain isolated indications while an integrated approach of both methods for flexible materials has not yet been conducted to the best of our knowledge. Therefore, we opted for a step-by-step approach with a first focus on the in-depth understanding of the underlying mechanism based on experimental data only.

Reviewer 3 Report

Comments and Suggestions for Authors

The subject is intriguing, yet a few adjustments are necessary before it can be published: The current references and literature review lack depth. It's advisable for the authors to incorporate more recent sources. Emphasize the originality of this research. Elaborate on justifying the results, ensuring validation through additional references.

Author Response

In Response to Reviewer #3

Overall Comments: The subject is intriguing, yet a few adjustments are necessary before it can be published: The current references and literature review lack depth. It's advisable for the authors to incorporate more recent sources. Emphasize the originality of this research. Elaborate on justifying the results, ensuring validation through additional references.

Response and action taken:

We thank the reviewer for his/her highly valuable comments. Indeed, the references seem out of date. This is due to the confined focus of the paper on hydrocarbon separations on flexible MOFs, something that has not yet been readily tackled as most publications focus on gas separations around CH4, CO2 and N2. However, we did amend the manuscript to your comments in order to a) provide a more actual update of the current scientific work in the literature review section and bring it into a greater context as well as b) to emphasize the originality of our research based on a).

Reviewer 4 Report

Comments and Suggestions for Authors

Manuscript of Glaser and co-workers describe the application of two flexible MOFs on gas separation. Although the topic of the manuscript is interesting, the manuscript is well written, experiments are well designed and performed, I believe that this work lacks of novelty, due to the overlap with other paper previously published (see ref. 33-35), also mentioned many times in the main text of this manuscript. Only section 3.3 seems to be novel. Thus, I believe that the novelty required by Nanomaterials is precluded. I suggest to submit to a more specialized journal

Author Response

In Response to Reviewer #4

Overall Comment: Manuscript of Glaser and co-workers describe the application of two flexible MOFs on gas separation. Although the topic of the manuscript is interesting, the manuscript is well written, experiments are well designed and performed, I believe that this work lacks of novelty, due to the overlap with other paper previously published (see ref. 33-35), also mentioned many times in the main text of this manuscript. Only section 3.3 seems to be novel. Thus, I believe that the novelty required by Nanomaterials is precluded. I suggest to submit to a more specialized journal.

Response and action taken:

We thank the reviewer for his/her comments. We agree that in terms of theoretical background, section 3.1 (for thermodynamic analysis based on the D-UAT) and section 3.2 (for kinetic analysis) rely heavily on already published papers. However, within these sections, the new theories are applied to a new set of sorptives which are physically very similar but chemically different (four C4-hydrocarbons) and thus provide new insights. We also agree that the largest novelty lies within section 3.3, where separation experiments are described in detail. With the application of a self-built hybrid apparatus incorporating both gravimetric as well as volumetric data-recording in combination with the insights gathered in 3.1 and 3.2, we were able to clearly identify the different mechanism at play and unveil a complex dependence of the overall selectivity on sorption enthalpy, rate of structural transition as well as steric effects.

By only studying the gas separation experiments alone without the pre-existing knowledge of the isolated effects during thermodynamic equilibrium as well as kinetic processes, this would not have been possible as herein, these effects are overlapped and inseparable.

While several research groups have published high sorption selectivities based on IAST-calculations and single-gas isotherms only, we strongly believe that this manuscripts in its entirety provides adequate novelty that is required for a publication in nanomaterials.

Round 2

Reviewer 2 Report

Comments and Suggestions for Authors

The revised version is acceptable for publication.

Comments on the Quality of English Language

The revised version is acceptable for publication.

Author Response

-

Reviewer 3 Report

Comments and Suggestions for Authors

Accept.

Author Response

-

Reviewer 4 Report

Comments and Suggestions for Authors

Considering my first report, authors do not changed manuscript or performed new experiments, thus my first opinion remains.

Author Response

Intro:

We thank the Reviewer again for the comment and appreciate you taking a stand in order to ensure a high quality within the renowned nanomaterials journal. However, we are still very much convinced that the broad readership of the journal will benefit from the novel findings reported in our manuscript. Below are our underlying thoughts to this claim.

Your remark:

In your opinion, chapter 3.3 is the only novel insight in the manuscript while chapters 3.1. and 3.2 are having a large “overlap with other paper previously published” while these former publications are “mentioned many times in the main text of this manuscript”.

Detailed description:

Within chapter 3.1 in our manuscript, the thermodynamic equilibria for the sorption of C4-molecules on the MOFs Cu-IHMe-pw and Cu-IHEt-pw are described in detail. Herein, a) we have found novel insights based on a new probing system with b) the application of a methodology that was reported before by our group.

Regarding a) - While, yes, the two MOFs were part of a previous study regarding the thermodynamics of the flexible materials where these were probed with n-alkanes, there were no insights gathered or described how chemically different sorptives would trigger the gate-opening of these frameworks. Therefore, we see a novelty within this manuscript as herein, the differences between the branched and linear olefins and paraffins can be derived and thus also the potential for a thermodynamically driven separation mechanism.

Regarding b) - What is furthermore applied in this manuscript is the stringent methodology to analyse the sorption equilibrium data, which was indeed taken from the previous publication. The following sentences are taken directly from our manuscript and only refer to the methodology and not the previous insights on the probing systems where the citations 44 and 45 belong to our previous work:

  • For the analysis of the equilibrium sorption isotherms of the studied systems, the Dubinin-approach [51,52] is used as done in previous studies for the MOF systems under investigation [44,45]
  • The specific boundaries of the structural transition, meaning the gate-opening ends for adsorption and the gate-closing start  for desorption, can be determined by applying the ESW-theory as described by Adolphs [53] and previous work [44].

Generally, the pattern of applying already established methodologies is state-of-the-art in this research community. Please consider the below stated publications regarding the separation of gases on MOFs.

  • https://doi.org/10.1038/nature11893 - 1863 citations
  • https://doi.org/10.1002/anie.201506345 - 206 citations
  • DOI: 10.1126/science.aaf6323 - 752 citations

All of these report MOFs that were published before but probe them with a new set of sorptives. All of these report sorption equilibrium data in the more established classical form with pressure vs. molar loading, which is equally a methodology as the Dubinin-approach. Within the last chapters, these publications report kinetic and gas mixture data based on established methods, similar to our work.

Equally to chapter 3.1, our chapter 3.2 only refers to methodologies reported in a previous work of ours while still providing novelty due to the new MOF-sorptive systems and again the insights on the differences between the branched and linear olefins and paraffins and how potential kinetic separation pathways could be established. All of these insights gathered in chapters 3.1 and 3.2 are necessary in order to understand in detail the mechanism switch from thermodynamic to kinetic and even size exclusion reported in chapter 3.3.

Therefore, we kindly ask you to reconsider your review for our manuscript and recognize the novelty of the entire work worthwhile for the renowned journal of nanomaterials.